# Prevalence and Risk Factors for Anxiety Symptoms among Student Nurses in Gauteng Province of South Africa

**DOI:** 10.3390/bs13080630

**Published:** 2023-07-28

**Authors:** Maleke Manana, Sam Thembelihle Ntuli, Kebogile Mokwena, Kgomotso Maaga

**Affiliations:** 1Department of Public Health, School of Health Care Sciences, Sefako Makgatho Health Sciences University, Pretoria 0204, South Africakebogile.mokwena@smu.ac.za (K.M.);; 2Department of Statistical Sciences, School of Science and Technology, Sefako Makgatho Health Sciences University, Pretoria 0204, South Africa

**Keywords:** anxiety, nursing students, Gauteng province, South Africa

## Abstract

**Background:** Globally, mental disorders are common among nursing students; therefore, effective prevention and early detection are urgently needed. However, the prevalence rate of anxiety symptoms has not been investigated in South African nursing colleges. **Aim:** The study aimed to assess the prevalence of anxiety symptoms and their sociodemographic risk factors among nursing students in Gauteng province, South Africa. **Methods:** This cross-sectional descriptive study was conducted at Chris Hani Baragwanath and SG Lourens nursing colleges in the first week of June 2022. A purposeful sampling technique selected the third- and fourth-year nursing students aged ≥ 18 years registered at the two nursing colleges. The seven-item Generalised Anxiety Disorder scale was used to assess anxiety symptoms. **Results:** The prevalence of anxiety symptoms was 74.7% (95% confidence interval: 69.9–78.9). Being a student at nursing college B, being in the fourth academic year of study and use of substances were identified as predictors of anxiety symptoms in these nursing students. **Conclusions:** The prevalence of anxiety symptoms in this study is relatively high, with predictors of developing anxiety being a student at nursing college B, in the fourth academic year and current use of psychoactive substances were predictors of anxiety symptoms. These findings highlight the need to develop interventions and strategies to promote mental health assessments and management to prevent and reduce the problem of mental disorders among nursing students.

## 1. Introduction

The nursing profession constitutes a fundamental part of the healthcare system and is critical in providing quality care and patients’ quality of life worldwide. This profession also offers various services, including health education and disease prevention and management [1]. It is one of the most stressful and emotionally draining occupations which often experiences a variety of stressors such as low wages, gender-based violence, poor working conditions, long work hours, time constraints, meeting patient’s needs, irregular schedules, and lack of professional support [2,3]. These may harm not only the quality of life and social functioning of the healthcare providers but also affect the outcome of the patients [4,5].

Mental disorders are a common phenomenon in nursing, and there is a growing concern about these disorders among university students. Anxiety is one of the most common mental health problems reported among nursing students [6,7,8,9]. Although anxiety has been linked to adverse health outcomes, little attention and resources are allocated to these disorders, especially in the student population. Anxiety disorders have been linked with reduced quality of life and premature death and are considered a risk factor for other comorbidities [10]. Specifically, for students, anxiety has also been linked with reduced academic performance and substance use, which heightens the risk of dropping out as well as addiction [11]. Studies demonstrated inconsistent findings concerning the prevalence of anxiety symptoms among nursing students, with many reporting rates below 40% [7,8,12,13,14], while others found prevalence rates above 40% [6,9,15,16]. Furthermore, many studies are showing that approximately 2–13% of nursing students experience severe anxiety symptoms [8,16,17,18].

Given the highly demanding nature of the nursing course, there are also various academic-related risk factors that predispose nursing students to high levels of anxiety. The nursing program consists of hectic schedules which consist of lecture attendance, submission of assessments, the combination of theory and clinical tasks, writing examinations, as well as giving oral presentations. The way the program is structured is time-consuming with high study demands and can be quite burdensome on students, leading to burnout, fatigue, and exhaustion, all of which may lead to negative mental health outcomes such as anxiety [19]. The clinical aspect of the program may also take an emotional and physical toll on students as they are expected to handle medical equipment, translate theory into practice, and have direct contact with patients that are sickly. These students are at a heightened risk for emotional exhaustion, compassion fatigue, as well as anxiety due to the constant exposure to traumatic events [19]. Additionally, the lack of familiarity to medical equipment, as well as limited clinical experience, may also instil fear and anxiety as students may be anxious and afraid of making mistakes [19,20]. Hence this population, requires close monitoring to ensure the retention of better mental health outcomes.

There are numerous academic-related factors that further predispose students to anxiety. The academic level of study [11,16,21,22,23,24,25,26], for example, has been associated with anxiety. Some authors found that first year students are at a more increased risk of anxiety [27]. These authors explain that the unfamiliarity of the curriculum, the sudden change in environment, and the introduction into tertiary learning places them at a higher risk compared to their older counterparts who have had time to gradually adapt to the curriculum [10]. However, others have found a positive association between higher years of study and anxiety [11]. Hwang and Kim [19] argued that students’ anxiety levels increase as workload increases during subsequent years of study and they are introduced into clinical work. The attempt to balance the two may be a stressor for senior students, and because there are no clinical requirements during the first year of study, junior students may not be affected as much. The curriculum embedded within the nursing course is rigorous, demanding, and time-consuming, which affects the social lives of students. Given the academic requirements, students may not have adequate time to socialize or entertain themselves, which makes them vulnerable to burnout, fatigue, stress, and, ultimately, anxiety [11]. It was found that students who did not make time for socialization were 50% more likely to be anxious than those who took a break at least once a week [11]. Unfortunately, some students may adopt maladaptive behaviours to help them cope with the curriculum. For example, the excessive use of substances has been identified as a widespread problem among the student nursing population [24].

Various demographic risk factors have also been associated with anxiety symptoms among nursing students, including age [21] and the female gender [9,16,22,23]. Female students have been identified to be at a higher risk for anxiety as compared to males and this is due to a variety of social and biological factors that affect men and women differently. For example, a study conducted by Li and colleagues [8] identified that in many households, women are more likely to do a large majority of the household chores, which may cause an imbalance between family, social, work, and academic responsibilities. This is especially true for female students that still live at home, where traditional gender roles are expected. Closely related to this is the stance that married students are at a heightened risk for anxiety [7] as opposed to those who are single. Similar to the imbalance experienced by females in traditional households, married students bear a lot of family responsibilities that may interfere with their academics. For example, Wathelet and colleagues [23] found that students with children were more likely to experience stress than those who did not have children. However, an opposing school of thought maintains that single students are more likely to be anxious [16] due to the lack of immediate support that comes with having a partner.

Family financial status [8,9,23] has also been associated with anxiety, as it has been displayed that nursing students from disadvantaged backgrounds are at an increased risk. Additionally, the role of COVID-19 played a significant role in worsening the mental health of the nursing student population, as high rates of anxiety have been reported since the emergence of the pandemic [9,21]. It was identified that fear of becoming infected as well as the lack of sufficient personal protection equipment [16] were reported as risk factors for heightened levels of anxiety among nursing student populations. Moreover, the uncertainty brought forth by the pandemic due to school closures that brought interruptions and changes to the academic program further worsened the mental health of students. Despite the high rates of anxiety symptoms and their risk factors, there is a lack of information concerning the status of anxiety symptoms and their associated risk factors in South African nursing colleges. Therefore, this study determines the prevalence of anxiety symptoms and demographic risk factors associated with anxiety among students at the two nursing colleges in Gauteng province, South Africa.

## 2. Methods

### 2.1. Study Design and Setting

This was a cross-sectional study conducted at Gauteng province nursing colleges. The two colleges offer a four-year undergraduate diploma in nursing according to the South African Nursing Council regulation 425 of 22 February 1985. The researchers completed the data collection within the first two weeks of June 2022.

### 2.2. Study Population, Sampling Technique, and Sample Size

The study population comprised 3rd and 4th year nursing students, aged ≥ 18, who were registered at the two nursing colleges. The reason for selecting the 3rd and 4th year nursing students is because they have had an extended period as nursing students, and the 1st and 2nd year students have had a short time. The collective student intake from both nursing colleges was 1320; thus, based on this number, a total sample size of 298 was calculated using Roasoft calculator at a 95% confidence interval, a 5% margin of error, and a 50% response distribution. Thus, the researchers used a purposive sampling technique to select the study participants undergoing their 3rd and 4th year, after which, all participants were recruited for participation. The students were gathered in a lecture hall and informed about the study’s aim and were provided with ample time to decide whether or not they wanted to participate. Only those who were interested and willing to participate in the study completed the consent form and questionnaire. A total of three hundred and seventy-nine (379) nursing students participated in this study.

### 2.3. Data Collection

The researchers developed a self-administered questionnaire using literature pertinent to collect the data. The tool collected the nursing student’s demographic data, including age, gender, marital status, place of residence, academic years of study, and substance use. The Generalised Anxiety Disorder item-7 (GAD-7), a screening tool widely used to evaluate anxiety symptoms, was used [28]. The GAD-7 questionnaire consists of 7 items, scored on a four-point Likert scale ranging from 0 to 3, in which 0 means not at all and 3 means nearly every day. The scores of the items were summed up and categorized into four groups: 0–4 indicates no anxiety, 5–9 indicates mild anxiety, 10–14 indicates moderate anxiety, and a score equal to or above 15 indicates severe anxiety. In the current study, Cronbach’s *α* was 0.82.

### 2.4. Data Analysis

The data were captured using Microsoft Excel 2016 (Microsoft Corporation, Redmond, Washington, DC, USA) and imported to STATA version 17 (StataCorp., College Station, TX, USA) for analysis. The percentages and numbers were used to interpret categorical data, while the mean and standard deviation were used to analyse continuous variables. The study used logistic regression to determine demographic factors associated with anxiety symptoms. All the variables with a *p*-value of less than 0.25 in a bivariate logistic regression analysis were considered statistically significant [29,30,31]. In multivariable logistic regression analysis, variables with a *p*-value < 0.05 were considered statistically significant. The Hosmer–Lemeshow goodness-of-fit test was used for model fitness and was found to be χ^2^ = 6.98; *p* = 0.3231 for the multivariable logistic regression model. The multicollinearity of independent variables was assessed using the variance inflation factor and found to be <5, which indicates that the independent variables are not linear combinations of each other.

### 2.5. Ethical Approval

The study obtained ethical approval from Sefako Makgatho University Research Ethics Committee (Ref: SMUREC/H/38/2021). Permission to conduct the study was obtained from the Gauteng provincial Department of Health (Ref: GP 202109 089) and the principals of each nursing college. The researchers fully informed the students about the study objectives; participation was voluntary and assured anonymity. All the students who agreed to participate completed the informed consent before completing the questionnaire.

## 3. Results

### 3.1. Demographic Characteristics

A total of 379 student nurses participated in the study. Their mean age was 30.6 (SD: 7.9), ranging from 22 to 58 years. The majority (*n* = 228, 60.2%) of the participants were less than 30 years of age. A large proportion (*n* = 327, 86.3%) were females, and 73% (*n* = 275) were single. Nearly all (*n* = 377, 99.47%) were African, and two (0.53%) were of other races.

More than two-thirds (*n* = 272, 71.8%) of the participants reside in Gauteng province, (*n* = 207, 54.9%) were from nursing college B, and (*n* = 265, 69.9%) were in the fourth academic year of study. Slightly more than half (*n* = 200, 52.8%) of the participants were current substance users. Of these (*n* = 247), 76.92% and 15.79% were drinking alcohol and smoking cigarettes, respectively. Few (4.75%) of the substance users used illicit drugs. A detailed description of the demographic characteristics of the participants is shown in Table 1.

Overall, of the nursing students who answered the question on whether the college has an office to consult when experiencing mental stress, 85% (*n* = 323) indicated that they have an office to consult when experiencing mental stress. There was no statistically significant difference between nursing colleges A and B concerning the availability of an office to consult when nursing students experienced mental disorders (84.3% versus 86.4%; *p* = 0.563). Few (11.6%) participants had consulted about mental health in the past six months. No statistically significant difference was observed in consults about mental health in the past six months between nursing colleges A and B (14.6% versus 9.2; *p* = 0.101).

### 3.2. Prevalence and Factors Associated with Anxiety among Nursing Students

The majority, 74.7% (95% CI: 69.9%; 78.9%), of the nursing students had anxiety symptoms. Figure 1 illustrates that 55.2% of the nursing students had mild to moderate and 19.5% had severe anxiety symptoms.

As shown in Table 2, in the univariate logistic regression analysis, participants aged 50 years and older (OR: 3.9; 95% CI: 0.49; 30.7), unmarried (OR: 1.9; 95% CI: 0.81; 4.25), in nursing college B (OR: 1.5; 95% CI: 0.95; 2.42), in the fourth academic year of study (OR: 2.6; 95% CI: 1.60; 4.24), and currently using substances (OR: 1.8; 95% CI: 1.15; 2.93) were identified as predictors of anxiety symptoms (*p* < 0.25).

In multivariate logistic regression analysis, participants in nursing college B (OR: 2.0; 95% CI: 1.21; 3.36), in the 4th academic year of study (OR: 3.3; 95% CI: 1.90; 5.80), and currently using substances (OR: 2.7; 95% CI: 1.55; 4.58) were found to be significantly associated with anxiety symptoms.

## 4. Discussion

The current sample was predominately single and below the age of 30, which is consistent with the average age and marital status of undergraduate students. Similar demographics were found in other studies as well [6]. The nursing profession is predominately female, which explains why the sample in the current study consisted largely of female participants [9,16,22,23]. This study evaluated anxiety symptoms and their associated risk factors among nursing students at Nursing Colleges in Gauteng province, South Africa. A prevalence rate of 74.7% was found in the current study, which is comparable to the rate of 70.6% reported among nursing students in Jordan [9]. The findings of this study are higher than the rates of 6.2% reported in Iran [7], 62.9% in Brazil [6], 38.9% in Turkey [13], 42.8% in Israel [16], 30.5% in Japan [14], and the rate between 30% and 45% in China [8,12,15].

The high prevalence of anxiety in the current study may be due to numerous socio-demographic and academic factors, which may differ across continents and countries. The other possible explanation is that the expectation of completing clinical work in a healthcare system that has a high volume of patients with limited resources could predispose one to potential mental disorders [32,33]. As an illustration, the study found that nursing students attending College B were at a two-fold risk of exhibiting anxiety symptoms as compared to those in College A. College B is situated in the biggest township in South Africa, which means nursing students may be exposed to more patients in the midst of limited resources during their clinical work. College A, on the other hand, is situated in a smaller township, which may indicate a lower volume of patients. Additionally, although both colleges are situated in Gauteng, which is highly crime-ridden, College B is located in Johannesburg, which has a higher murder rate than Tshwane [34], where the other college is located. This implies that the students in College B may have higher rates of anxiety due to the greater exposure to crime, as studies have shown that students living in places where crime is rampant are more likely to be anxious [35]. This shows the impact of the different factors within the nursing colleges that influence the mental health status of the students. Unfortunately, due to the cross-sectional nature of the study, the differences could not be explored in greater depth.

Similar to other studies on anxiety [16], the current study found that the majority (55.2%) of the students exhibited mild–moderate symptoms of anxiety, whereas fewer students (19.5%) reported severe symptoms. A cross-sectional study reported that 1.69% [8] and 13.1% [16] of nursing students had severe anxiety symptoms during the COVID-19 pandemic lockdown, while other studies reported rates of 5.8% [17] and 8.3% [18] of students suffering from severe anxiety before the COVID-19 pandemic started. The high prevalence and severity of anxiety symptoms in this sample is very concerning, considering the low utilization of mental health services, despite a widespread access. In our study, most (*n* = 323; 85%) participants reported having an office at the college to consult when experiencing mental disorders. However, few (11.6%) had consulted a professional in this office concerning their mental health. Several studies have reported that students were less likely to seek help from student counselling centres when experiencing mental disorders [36,37]. In the present study, the challenges experienced by students in avoiding seeking professional help when experiencing mental disorders are unclear. However, other studies reported fear of documentation on academic records, fear of unwanted intervention, and lack of time as the main reasons for not seeking help [38]. Given that mental health may affect individual aspects of life, training institutions should implement different strategies, including psychological support [33] and self-compassion training [17], to alleviate anxiety.

There was a combination of academic and socio-demographic factors that were significantly associated with anxiety symptoms. For example, while other studies found no association between the academic level of study and the development of anxiety symptoms [10,12,16], the present study findings found that the academic study level of students predicted anxiety symptoms. These results are synonymous with other studies in the field [12,22,25,27]. It was identified that fourth-year nursing students had a 3-fold increased risk of anxiety symptoms as compared to third-year students. It is explained that anxiety increases with academic progression due to the increased academic workload that has to be balanced with clinical requirements [19]. Contrary arguments state that senior students are less likely to become anxious due to their ability to gradually adapt to the nursing curriculum as compared to first year students that are unfamiliar with the tertiary environment and learning content [26]. However, there is an increased pressure, especially in the final year of study, for students to perform well academically in order to meet graduation requirements, which may induce feelings of anxiety. Additionally, there is also an added component of entering the job market, which can also heighten anxiety levels among final year students [19].

Regarding the socio-demographic predictors of anxiety symptoms, some studies found that age was a predictor of the occurrence of anxiety symptoms [18], while others found no association between age and anxiety symptoms [16]. In the current study, however, it was found that older students (≥50 years) were at a 3.9-fold increased risk for anxiety symptoms on bivariate but not multivariate logistic regression analysis. It may be reasoned that the higher levels of anxiety among older students may be due to various challenges faced by mature students pursuing undergraduate studies. For example, older students are more likely to have family responsibilities which may lead to difficulties in balancing academic, family, and work-related responsibilities, and hence lead to negative mental health outcomes. Moreover, this may also be explained by the COVID-19 pandemic, as older age was regarded as a risk-factor for infection [16]. As a result, the older students in the sample may have had more anxiety about the fear of becoming infected as compared to those that were younger.

The present study further found that gender did not predict the occurrence of anxiety symptoms, which has also been replicated in many other studies [21,26]. In contrast, other studies found that the female gender was a significant risk factor for anxiety symptoms [16,21,23]. Although marital status did not remain significant after multivariate analysis, it was significant at a univariate level. It was found that unmarried students (this includes single, widowed, and divorced) were twice more likely to be anxious than those who were married. An Iranian study conducted among nursing students supports this finding [7], however, other studies found no association between marital status and anxiety [16].

Previous studies found that family financial status is associated [8] and not associated [9] with anxiety symptoms. In this study, we did not evaluate the relationship between family financial status and anxiety, which needs further investigation. Some studies have revealed that nursing students with mental disorders are at risk for problems related to substance use [39,40]. There was a strong, positive correlation between substance use and anxiety, as students that used substances had a 2-fold increased risk of anxiety symptoms. The use of substances may be due to the highly stressful nature of the nursing course, as students may resort to using substances as a coping mechanism. The prevalence of substance use among this sample was concerning, as more than half (*n* = 247, 65.17) of the participants were current substance users, with 76.92% consuming alcohol, 15.79% smoking tobacco products, and 4.75% using illicit drugs, mainly dagga, cocaine, and heroin. Similar rates were identified among Australian nurses, with 92.5% and 18% consuming alcohol and smoking cigarettes, respectively [41]. This is an indication that there may be a serious substance use problem among students and professionals in the nursing industry, thus warranting further investigation. This also seems to be a common problem among university students as well. For example, a cross-sectional study conducted in four different public universities in the Eastern Cape province of South Africa found that the prevalent substances used by students were 43.1% tobacco products, 55.9% alcohol, and 57.2% cannabis [24].

It is therefore essential to increase efforts to address substance abuse at a tertiary level and further intensify the importance of seeking mental health services among nursing students. The high prevalence in this sample indicates that a majority of nursing students are at risk of anxiety and are largely undetected. This has detrimental consequences, first, on the direct physical, emotional, and academic lives of students, and secondly, on the patients under their care during clinical trials, as well as patients that will be under their care after they successfully enter the nursing profession. Thus, disregarding the mental health status of the student nurse population will have a negative impact on the overall health system.

## 5. Limitations

As with cross-sectional studies, the study could not establish a cause-and-effect relationship. Secondly, the study was conducted in one province, which limits the generalisation to other provinces of South Africa.

## 6. Conclusions

The study findings revealed that anxiety symptoms are exceedingly prevalent in the sample. Factors related to anxiety symptoms include the older students, being at college B, being in the fourth academic year of study, and the use of psychoactive substances. Anxiety that starts during student nursing days is likely to continue after qualifying, which suggests a high prevalence of anxiety among professional nurses.

## 7. Recommendations

With high prevalence of anxiety being reported across the board, this study recommends that because of their caring responsibilities, special attention should be paid to the mental health of nurses, because failing to do so compromises their ability to perform their duty and compromises the health of the patients under their care. It is also recommended that wellness programs for nursing students and other nurses should specifically focus on the mental aspect of their health, which should include custom-made mental health interventions for nurses and nursing students. Lastly, further research is needed to determine the prevalence of anxiety in other nursing colleges across the country.

## Figures and Tables

**Figure 1 behavsci-13-00630-f001:**
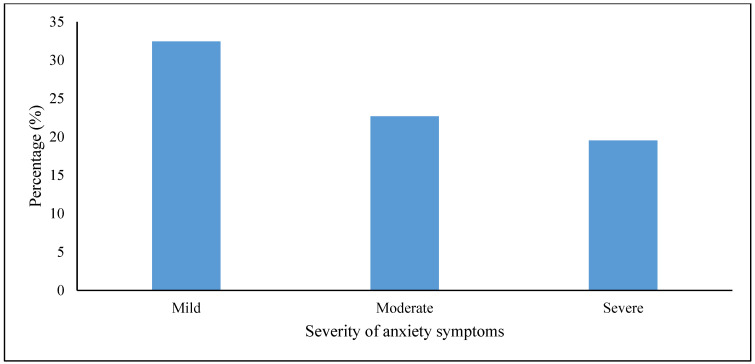
Severity of anxiety symptoms.

**Table 1 behavsci-13-00630-t001:** Demographic information of the participants, *n* = 379.

	No	%
Age (years)		
≤30	228	60.2
30–39	88	23.2
40–49	47	12.4
50+	13	3.4
Unspecified	3	0.8
Gender		
Male	52	13.7
Female	327	86.3
Marital Status		
Single	275	72.6
Married	69	18.2
Living with partner	27	7.1
Divorced/Widower	8	2.1
Place of residence		
Gauteng	272	71.8
Other provinces	106	28.0
Unspecified	1	0.3
Nursing College		
A	172	45.4
B	207	54.6
Academic years of study		
Third year	113	29.8
Fourth year	265	69.9
Unspecified	1	0.3
Office to consult when experiencing mental stress		
No	56	15.00
Yes	323	85.00
Consulted professional about mental health in the past six months		
No	44	11.60
Yes	335	88.40
Currently using substances		
No	131	34.56
Yes	247	65.17
Unspecified	1	0.2
Type of substance used (*n* = 247)		
Smoke tobacco products	39	15.79
Consuming alcohol	190	76.92
Illicit drugs used	18	4.75

Illicit drugs used: dagga, cocaine, heroin, etc.

**Table 2 behavsci-13-00630-t002:** Predictors of anxiety symptoms among nursing students.

	N	Anxiety Symptoms, *n* (%)		Univariate Logistical Regression	Multivariate Logistical Regression
		Yes	No	OR (95% CI)	OR (95% CI)
Age (years)		30.9 ± 8.28	29.8 ± 6.76		
<30	228	172 (75)	56 (25)	Ref	Ref
30–39	88	61 (69)	27 (31)	0.7 (0.43; 1.27)	0.9 (0.51; 1.67)
40–49	47	36 (77)	11 (23)	1.1 (0.51; 2.23)	1.6 (0.68; 3.53)
50+	13	12 (92)	1 (8)	3.9 (0.49; 30.7) *	6.4 (0.76; 53.61)
Gender					
Male	52	36 (69)	16 (31)	Ref	
Female	327	247 (75)	80 (25)	1.4 (0.72; 2.60)	
Marital Status					
Living with partner	27	17 (63)	10 (37)	Ref	Ref
Married	69	51 (74)	18 (26)	1.7 (0.65; 4.30)	1.3 (0.45; 3.98)
Unmarried	283	215 (76)	68 (24)	1.9 (0.81; 4.25) *	1.2 (0.48; 3.09)
Nursing College					
A	172	121 (70)	51 (30)	Ref	Ref
B	207	162 (78)	45 (22)	1.5 (0.95; 2.42) **	2.0 (1.21; 3.36) **
Academic year of study					
Third year	113	69 (61)	44 (39)	Ref	Ref
Fourth year	265	213 (80)	52 (20)	2.6 (1.60; 4.24) **	3.3 (1.90; 5.80) **
Substance use					
No	178	122 (69)	56 (31)	Ref	Ref
Yes	200	160 (80)	40 (20)	1.8 (1.15; 2.93) **	2.7 (1.55; 4.58) **

OR = odds ratio; CI = confidence interval; *p*-value < 0.25 *; *p* < 0.05 **. Unmarried: Single/Divorced/Widower.

## Data Availability

Data is available on request from the authors.

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
