# Peer review of "Prevalence and Risk Factors for Anxiety Symptoms among Student Nurses in Gauteng Province of South Africa"

_behavsci, 2023, doi:10.3390/bs13080630_

Round 1
Reviewer 1 Report
Thank you for the opportunity to review the manuscript. “Prevalence and Risk factors of Anxiety Symptoms among Student Nurses in Gauteng Province of South Africa”.
Background:
Abstract: Background: Globally, mental disorders are common among nursing students; therefore, effective prevention and early detection are urgently needed. However, the prevalence rate of anxiety symptoms has not been investigated in South African nursing colleges. Aim: The study aimed to assess the prevalence of anxiety symptoms and its sociodemographic risk factors among nursing 13 students in Gauteng Province, South Africa. Methods: This cross-sectional descriptive study was conducted at Chris Hani Baragwanath and SG Lourens nursing colleges in the first week of June 2022. A purposeful sampling technique selected the 3rd and 4th year nursing students aged ≥18 16 years registered at the two nursing colleges. The 7-item Generalized Anxiety Disorder scale was used to assess anxiety symptoms. Results: The prevalence of anxiety symptoms was 74.7% (95% 18 confidence interval: 69.9 - 78.9). Being a student at nursing college B, being in the 4th academic year of study and use of substances were significantly were ntified as predictors of anxiety symptoms in these nursing students. Conclusion: The prevalence of anxiety symptoms in this study is relatively high, with predictors of developing anxiety being a student at nursing college B, in the 4th academic year and current use of psychoactive substances were predictors of anxiety symptoms. These findings highlight the need to develop interventions and strategies to promote mental health assessments and management to prevent and reduce the problem of mental disorders among nursing students.
Keywords: Anxiety; Nursing students; Gauteng Province; South Africa
It is a study with citation potential, with a delimited methodology whose results are expressive for public health and for nursing. In the introduction, the authors present an important overview of the reality of mental illness among nursing students and relate this phenomenon to factors related to nursing education itself, which in itself already points to the emergence of common mental disorders such as anxiety.
The method strictly follows the standards established by STROBE, however, it is still necessary to present the sample calculation.
The descriptive analysis presented is correct, however, table 2 presents variables whose sum of participants does not reach the number presented by the authors (eg Age and substance use). Was there a loss of participants or did some not answer all questions in the instrument?
The discussion is relevant and presents many articulations with similar international studies.
The references are current and relevant to the construction presented, however the percentage of references with more than five years reaches almost 30%.
Reviewer 2 Report
- Section 2.2: How many students were identified as a potential candidate for the study and were explained the aims in the lecture hall? How many students refused to participate in the study? Were they given enough time to think about the study?
- Section 2.3: GAD-7 value 0 should mean "Not at all" instead of "Not sure at all".
- The Type I error rate is 20% (p value 0.2) which is quite higher than the usual error rate of 5%. Any reason why the author selected this high error rate in bivariate logistic regression model?
- "Overall, of the nursing student who answered the question on whether the college has an office to consult when experiencing mental stress (n=378, 85.5%) indicated that they have an office to consult when experiencing mental stress (data not shown)." Can you calrify this statement? Also, n = 378 which should be more than 99% rather than 85.5% based on your total N = 379.
- In the result section 3.2, the author identified predictors of anxiety based on univariate logistic regression analysis with the p-value of < 0.25. However, in the mehtod section, it was indicated that bivariate logistic regression will be assessed using the p-value 0.20. This is a discrepancy. Also, when you are using p-value of 0.2 or 0.25, range of confidence interval changes. 95% confidence interval makes sense if you are using your p-value 0.05.
- spelling mistkae "ntified" in result sectio of abstract.
- spelling mistkae "Thes" in the introduction section.
- grammer error "women are carry a disproportionate amount of household chores..." in introduction section.
- spelling mistkae "ha.d consulted" in discussion section.
- spelling mistake "nursingng students" in recommendations section.
Round 2
Reviewer 2 Report
Thank you for working on the comments. I congratulate and thank you for your contribution to the field.